# FAITH: FACTUALITY ALIGNMENT THROUGH INTEGRATING TRUSTWORTHINESS AND HONESTNESS

## ABSTRACT

Large Language Models (LLMs) can generate factually inaccurate content even if they have corresponding knowledge, which critically undermines their reliability. Existing approaches attempt to mitigate this by incorporating uncertainty in QA prompt during training, but these numerical scores lack the semantic richness for LLM to properly understand its internal states of trustworthiness and honestness, leading to insufficient factuality alignment. We introduce **FAITH** (Factuality Alignment through Integrating Trustworthiness and Honestness), a post-training framework for factuality alignment that integrates natural-language uncertainty signals with external knowledge. Specifically, we augment training datasets by computing confidence scores and semantic entropy from LLM outputs and mapping them into a knowledge state quadrant that describes the model's internal knowledge possession (trustworthiness) and answering behaviors (honestness) in natural language. Based on this enhanced data, we design a reward function that considers both correctness and uncertainty signals, and fine-tune the LLM using the Proximal Policy Optimization (PPO) algorithm. To further mitigate weakly grounded responses, we design a retrieval-augmented module that retrieves relevant external passages, improving the consistency between internal and external knowledge representations. Extensive experiments on four knowledge-intensive benchmarks demonstrate that FAITH enhances the factual accuracy and truthfulness of LLMs.

## 1 INTRODUCTION

Large Language Models (LLMs), such as GPT-4 (OpenAI, 2023), Llama3 (Dubey et al., 2024) and DeepSeek-v3 (DeepSeek-AI et al., 2024), have demonstrated impressive performance across a broad range of natural language processing tasks. Despite these advances, growing evidence shows that LLMs may generate outputs that are fluent but factually incorrect or fabricated, a phenomenon commonly known as hallucination (Huang et al., 2021; Ji et al., 2023). Such hallucinations pose substantial risks in knowledge-intensive and high-stakes domains, including legal, educational, and clinical applications (Alkaissi & McFarlane, 2023; Wang et al., 2023b).

A particularly concerning type of hallucination emerges when the model possesses the necessary knowledge but fails to articulate it accurately. This disconnect between internal knowledge and external expression, often termed the *know–tell* gap (Saunders et al., 2022; Li et al., 2025), not only undermines the model's ability to convey truthful information but also manifests as inconsistency of factual expression, where the model may produce an incorrect response in one instance yet a correct one in another (Manakul et al., 2023; Wang et al., 2023a).

In this work, we propose to post-train LLMs for enhancing factuality. Our study identifies several limitations in recent endeavors (Tian et al., 2024; Tao et al., 2024; Xue et al., 2025; Sun et al., 2025): (1) while these work introduce uncertainty for factual alignment, they directly use the numerical values into question-answering prompts during training, which lack semantic richness and are difficult for LLMs to understand and exploit for factuality-aligned expression; (2) they employ binary reward function in policy training, which simply focuses on whether the response is correct or not while ignoring to consider the confidence of LLM's response (*i.e.,* uncertainty), potentially encouraging guessing; and (3) they neglect the use of external knowledge, leaving potentially incorrect responses unrectified.

To address these limitations, we introduce FAITH (Factuality Alignment through Integrating Trustworthiness and Honesty), a post-training framework designed for factuality alignment in LLMs. FAITH incorporates three key designs: (1) When augment in-domain training datasets, beyond estimating the uncertainties (via consistency and semantic entropy) of LLMs for each question in datasets, we map these numerical values into a knowledge state quadrant (Liang et al., 2024) where each knowledge state is expressed in natural language and defined along two dimensions: *knowledge possession* (trustworthiness) and *answering behavior* (honesty). Unlike opaque numerical uncertainty values, incorporating knowledge states into QA prompt during training provides LLMs with semantically rich and interpretable guidance. (2) For policy optimization, we design a fine-grained reward function that consider both the correctness of response and LLM's uncertainty, providing more informative feedback than a binary reward, encouraging the policy model to align its outputs with their knowledge states. (3) To further improve reliability, we construct a vector database over the Wikipedia corpus (Karpukhin et al., 2020) and train a RAG model that retrieves external knowledge from the database as contextual input to rectify potentially incorrect responses generated by the policy model.

Through FAITH, we enhance LLM factuality in terms of precision and truthfulness. Extensive experiments show that FAITH consistently outperforms five recent strong baselines on three in-domain and one out-of-domain dataset. For example, on Llama3-8B, FAITH achieves 74.26% precision and 45.73% truthfulness on in-domain datasets, and 67.99% precision and 34.03% truthfulness on the out-of-domain dataset. Similar gains are observed on Mistral-7B-v0.1, demonstrating that FAITH's effectiveness generalizes across both models and datasets.

In summary, our contributions are as follows:

1. We introduce FAITH, a novel post-training framework for factuality alignment. FAITH advances the factuality alignment by its semantically rich knowledge state quadrant, fine-grained reward function, and employing external knowledge to ground LLM's response.

2. We conduct extensive experiments demonstrating that FAITH consistently outperforms strong baselines, with performance gains generalizing across multiple datasets and models. Meanwhile, we provide ablations to assess the contribution of each component.

3. We provide in-depth analyses of FAITH, including the impact of different knowledge state estimation strategies on inference performance and training-time scaling behavior with varying numbers of sampled responses $K$.

## 2 RELATED WORK

**Factuality Alignment.** To enhance LLM factuality, prior work has explored training-free strategies such as external knowledge augmentation (Kandpal et al., 2023; Jiang et al., 2023b), decoding methods (Chuang et al., 2024), and self-consistency techniques grounded in uncertainty estimation (Kadavath et al., 2022; Tian et al., 2023). More recently, post-training approaches, including SFT and policy optimization, have been applied to further improve factuality (Tian et al., 2024; Tao et al., 2024; Sun et al., 2025; Xu et al., 2024; Xue et al., 2025; Chen et al., 2025). Our work falls into this category, where we propose to map numerical uncertainty values into natural-language knowledge states and design a new reward function to enhance the model's expression of its knowledge.

**Uncertainty Estimation.** Uncertainty Estimation (UE) has long been studied in machine learning domain, including NLP, with vast majority of previous work focusing on discriminative tasks, such as sentiment analysis (Xiao et al., 2022). Specifically, the entropy of the predictive posterior and the negative predictive posterior probability of the most probable answer are used to quantify uncertainty in predictions (Lakshminarayanan et al., 2017; Bakman et al., 2024). However, LLMs pose new challenges for uncertainty estimation due to their generative paradigm. Recent studies have extended UE to generative models, introducing heuristic or probabilistic metrics such as entropy-based scoring (Malinin & Gales, 2021), semantic entropy that accounts for meaning equivalence in multiple generations (Kuhn et al., 2023), and similarity-based methods applied in tasks like machine translation (Fomicheva et al., 2020; Lin et al., 2022b). Other works explore black-box UE by leveraging sampled outputs (Chen & Mueller, 2024; Manakul et al., 2023), or prompt-based approaches where models verbalize their own confidence (Kadavath et al., 2022). Training-based methods have also been proposed to enhance linguistic self-assessments of uncertainty (Lin et al., 2022a).

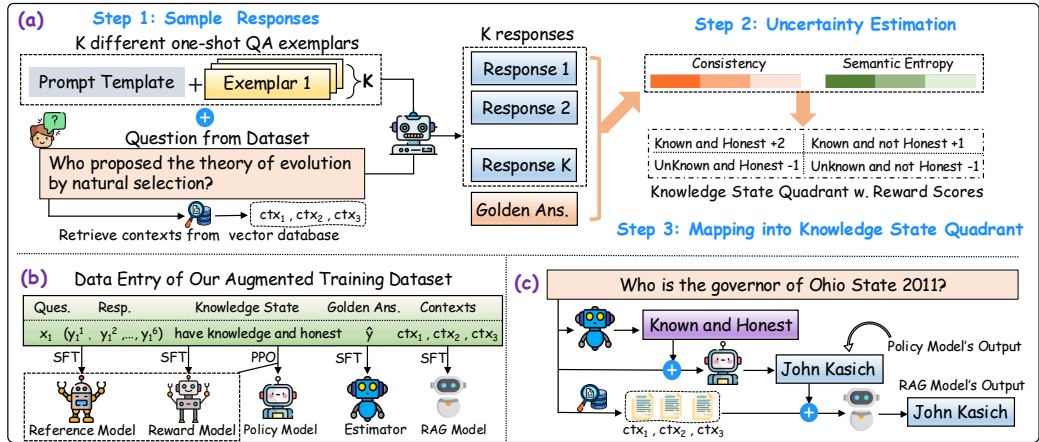

Figure 1: Illustration of the FAITH framework. Panel (a) shows the procedure for augmenting the training datasets, Panel (b) depicts model training with the augmented datasets, and Panel (c) presents the inference pipeline.

**Retrieval-Augmented Generation.** Retrieval-Augmented Generation (RAG) has become a widely adopted strategy for mitigating the limitations of parametric knowledge in LLMs by grounding responses in external evidence (Guu et al., 2020; Izacard & Grave, 2021; Zhong et al., 2023). However, recent studies reveal that contradictions may arise when retrieved knowledge conflicts with internal representations (Mei et al., 2024; Ni et al., 2024). To mitigate this, retrieval-based methods increasingly leverage structured repositories such as knowledge graphs for more reliable grounding (Baek et al., 2023; Zhang et al., 2023). Other approaches enhance factuality by incorporating retrieval into inference-time interventions, including memory-augmented architectures (Li et al., 2022) and entity-level embedding integration (Kang et al., 2022; de Jong et al., 2022). Generally, RAG provides a scalable way to continuously integrate knowledge, offering advantages over task-specific parameter editing.

## 3 PRELIMINARY

**Problem Definition.** We consider a standard *open-domain, closed-book QA* setting, where a language model $LLM$ is given a factual question $q$ and generates a short-form answer $a \sim LLM(\cdot \mid q)$ with probability $LLM(a \mid q)$. The answer is expected to be concise and factually correct, but in practice may suffer from failure of factuality due to uneven knowledge possession and the gap between knowledge and expression, while the autoregressive generation paradigm inevitably produces responses by sampling from token distributions.

Table 1: Categories of model output for QA.

| Abbr. | Explanation |
|-------|-------------|
| **KC** | Known and answered correctly |
| **KI** | Known but answered incorrectly |
| **KR** | Known but refused to answer |
| **UC** | Unknown but answered correctly |
| **UI** | Unknown but answered incorrectly |
| **UR** | Unknown and refused to answer |

Following prior work (Xue et al., 2025), we categorize model outputs into six types based on knowledge possession and response correctness, as summarized in Table 1. The research scope of this work is to encourage more instances of **KC** and **UR** (See § 5.1 for evaluation metrics).

## 4 METHOD

As shown in Figure 1, we introduce the framework of FAITH which enhances factuality alignment for LLMs. Specifically, we first augment three QA training datasets by estimating the uncertainties of LLMs for each question in these datasets and translating the numerical uncertainty values into natural-language descriptions drawn from the knowledge state quadrant (described in § 4.1). We then apply Proximal Policy Optimization (PPO) to finetune LLMs on the augmented datasets,

guiding them to answer questions combining their knowledge state; meanwhile, to mitigate internal knowledge insufficiency for confidently expressing knowledge, we train a RAG model to rectify potential incorrect answers produced by policy model (described in § 4.2). Finally, we conduct the inference in § 4.3.

### 4.1 TRAINING DATASET AUGMENTATION

As shown in Figure 1, Panel (a), we first sample responses to each question in in-domain QA datasets and then conduct uncertainty estimation, which is consistent with existing works of uncertainty estimation (Xiong et al., 2024; Kuhn et al., 2023; Aichberger et al., 2025; Kang et al., 2025). We do this for two purposes: (1) to measure the knowledge boundary of an LLM and locate whether a given question is within it; (2) to evaluate the honestness of an LLM when answering the given question.

**Sampling Responses from Training Datasets.** We sample responses from the training split of NQ-Open (Kwiatkowski et al., 2019), SciQ (Welbl et al., 2017b) and TriviaQA (Joshi et al., 2017) datasets, and each dataset $\mathcal{D}$ contains a set of $N$ question-answer pairs $\{(x_i, \hat{y}_i)\}_{i=1}^N$, where $x_i$ and $\hat{y}_i$ represent the $i$-th question and golden answer in $\mathcal{D}$, respectively. Specifically, for each question $q_i \in \mathcal{D}$, we prompt $q_i$ with $K$ different one-shot exemplars and obtain $K$ responses, denoted as $Y_i = \{y_i^k\}_{k=1}^K$. We set $K = 6$ in the main experiment, same as in baseline UAlign (Xue et al., 2025). Also, for fair comparison to baselines, we adopt the same temperature `T=0.2` with UAlign and also randomly preserve half of data entries in NQ-Open and TriviaQA datasets while preserve all entries in SciQ dataset.

**Uncertainty Estimation.** For the uncertainty estimation of generative LLMs, we employ consistency and semantic entropy. Consistency serves as an accuracy-based measure of confidence, and it reflects the accuracy of the generated $K$ candidate responses (Xiong et al., 2024), computed as follows:

$$Consistency(x_i) = \frac{1}{K} \sum_{i=1}^K \mathbb{1} \left\{ y_i^k = \hat{y}_i \right\}. \tag{1}$$

The semantic entropy (SE), on the other hand, captures uncertainty from the semantic dispersion of generated answers, determining the likelihood of each meaning $c$ rather than each generated sequence $y_i^k$ (Kuhn et al., 2023). It addresses the limitations of prior approaches, which are often affected by response length or by semantically identical answers expressed in different surface forms. Semantic entropy is defined as:

$$SE(x_i) = - \sum_c p(c \mid x_i) \log p(c \mid x_i) = - \sum_c \left( \left( \sum_{y_i^k \in c} p(y_i^k \mid x_i) \right) \log \left[ \sum_{y_i^k \in c} p(y_i^k \mid x_i) \right] \right). \tag{2}$$

By now, we augment each data entry in $\mathcal{D}$ from $(x_i, \hat{y}_i)$ to $(x_i, \hat{y}_i, Y_i, Consistency(x_i), SE(x_i))$. However, we argue that the numerical values of uncertainty, which will be used in QA training prompt (*e.g.,* "`Who starred in an officer and a gentleman ### Conf: 0.833 ### Entro: -0.`" in Xue et al. (2025)), cannot effectively guide LLM to understand and recognize its knowledge boundary, as raw numbers lack semantic meaning.

**Knowledge State Mapping.** To this end, we map consistency scores and semantic entropy onto our defined knowledge state quadrant (described below), thus rendering otherwise opaque numerical uncertainty values in semantically rich natural language descriptions. Specifically, we describe knowledge states of LLM based on two factors: *knowledge possession* and *answer behavior*, and this results in a quadrant with four knowledge states expressed in natural language: (i) **Have knowledge and honesty (KH)**, (ii) **Have knowledge but not honesty (K¬H)**, (iii) **Not have knowledge but honesty (¬KH)**, and (iv) **Not have knowledge and not honesty (¬K¬H)**.

We quantify the knowledge possession of LLMs for a given question $x_i$ through consistency defined in Equation 1, and the indicator function $\mathbb{1}$ is *Positive-Recall Exact Match (PREM)*, where $\text{PREM}(y_i^k, \hat{y}_i) = 1$ if $y_i \in \hat{y}_i^k \vee \hat{y}_i^k \in y_i$, otherwise $\text{PREM}(y_i^k, \hat{y}_i) = 0$, which is widely-used in short-form QA. On the other hand, we model the answer behavior of LLMs via semantic entropy.

Overall, the procedure for mapping consistency and semantic entropy into the knowledge state quadrant $\mathcal{S}$ is defined as follows (We explain the details of knowledge states in Appendix A.1.):

$$
s_i = \text{KnowledgeState}(x_i) = \begin{cases} \text{KH}, & \text{if } Consistency(x_i) > 0 \text{ and } SE(x_i) = 0, \\ \text{K$\neg$H}, & \text{if } Consistency(x_i) > 0 \text{ and } SE(x_i) \neq 0, \\ \neg\text{KH}, & \text{if } Consistency(x_i) = 0 \text{ and } SE(x_i) = 0, \\ \neg\text{K$\neg$H}, & \text{otherwise.} \end{cases} \tag{3}
$$

The knowledge state formulation (Eq. 3) allows us to characterize LLMs in terms of trustworthiness (knowledge posession) and honesty (answer behavior), and we finally augment each data entry from $(x_i, \hat{y}_i)$ to $(x_i, \hat{y}_i, Y_i, s_i)$.

## 4.2 TRAINING STAGE OF FAITH

We aim to leverage both internal and external knowledge to enhance the expression of existing knowledge, bridging the gap between knowing and telling. To this end, as shown in Figure 1, Panel (b), we first train a policy model using PPO to align LLM's responses with its internal knowledge states. We then train a RAG model to correct potentially incorrect responses by incorporating external knowledge. Finally, we introduce a knowledge state estimator that eliminates the need for sampling multiple responses during inference, thereby improving efficiency. All prompt templates used in training stage are provided in Appendix A.2.

**Reference Model Training.** We start from a pretrained base model and obtain a reference model $\pi_\mu$ through supervised fine-tuning (SFT). Specifically, the model is fine-tuned using pairs of the form $(\texttt{prompt}(x_i, s_i); \hat{y}_i)$, where the prompt incorporates both the question $x_i$ and the knowledge state $s_i$, and $\hat{y}_i$ is the golden answer. By fine-tuning the base model with these curated input–output pairs, the reference model establishes a foundation for subsequent policy optimization.

**Reward Model Training.** To align generation with knowledge state, we train a reward model with parameter $\theta$ to evaluate the generated response combined with the knowledge state. Different from existing binary reward $r_i \in \{0, 1\}$ which only focuses on whether the response is correct or not and ignores how confident the correct response is (Yao et al., 2025; Kirichenko et al., 2025; Xue et al., 2025), we propose a fine-grained reward function to focus on both the correctness of response and the uncertainty. Specifically, we propose a combined reward function:

$$
\begin{aligned} R_{\text{FAITH}}\left(x_i, y_i^k, \hat{y}_i, s_i\right) &= R_{\text{correctness}}\left(y_i^k, \hat{y}_i\right) + R_{\text{uncertainty}}\left(s_i\right) \\ &= \mathbb{1}_{y_i^k \equiv \hat{y}_i} + R_{\text{uncertainty}}\left(s_i\right), \end{aligned} \tag{4}
$$

where $s_i = \text{KnowledgeState}(x_i) \in \mathcal{S}$ and the $R_{\text{uncertainty}} \in \{+2, +1, -1, -2\}$ is defined by the following rules in terms of its knowledge state $s_i$ in $\mathcal{S}$:

$$
+2 \to \text{KH}, \qquad +1 \to \text{K$\neg$H}, \qquad -1 \to \neg\text{KH}, \qquad -2 \to \neg\text{K$\neg$H}.
$$

We parameterize the reward function into a reward model $RM_\theta$. Specifically, given a dataset $\mathcal{D}$ containing multiple tuples $(x_i, y_i^k, s_i, r_i^k)$, where $r_i^k$ is the reward value, the reward model minimizes the multi-class cross-entropy:

$$
\mathcal{L}_\theta = -\mathbb{E}_{(x_i, y_i^k, s_i, r_i^k) \sim \mathcal{D}} \left[\log p_\theta(r_i^k \mid x_i, y_i^k, s_i)\right]. \tag{5}
$$

Our fine-grained reward model provides more informative feedback than a binary reward, encouraging the policy model to align its generated responses with their knowledge state, where uncertainty is expressed in natural language form rather than numeric scores.

**Policy Model Training** Similar to reinforcement learning from human feedback (RLHF) (Ouyang et al., 2022), we employ PPO with a KL-divergence penalty to optimize LLMs for factuality alignment. Specifically, given a question $x_i$ paired with its knowledge state $s_i$, both the reference model $\pi_\mu$ and the policy model $\pi_\phi$ generate responses, while the reward model $RM_\theta$ evaluates the factual

reliability of a generated response $\tilde{y}_i$ in terms of correctness and uncertainty (i.e., knowledge state). The training objective is to optimize $\pi_\phi$ to maximize the expected reward:

$$\arg\max_{\pi_\phi} \mathbb{E}_{x\sim\mathcal{D},\ s\sim\text{KnowledgeState}(x),\ \tilde{y}\sim\pi_\phi(x,s)} \left[ \underbrace{RM_\theta(x,\tilde{y},s)}_{\text{reward}} - \beta \underbrace{\text{KL}\left[\pi_\mu(x)\| \pi_\phi(x,s)\right]}_{\text{penalty}} \right]. \quad (6)$$

**RAG Model Training.** We train a RAG model $\pi_{rag}$ to leverage external knowledge to rectify potentially incorrect answers produced by the policy model. To this end, we first build a vector database over the Wikipedia corpus (Karpukhin et al., 2020) using the BAAI General Embedding model[1]. We employ IndexIVFPQ in Facebook AI Similarity Search (FAISS) (Johnson et al., 2021) as the retriever to perform similarity search. For each question $x_i \in \mathcal{D}$, the retriever returns the top-3 most semantically relevant passages, denoted as $ctx_i = \{context_i^j\}_{j=1}^3$, which are used as context in prompt. Accordingly, we augment the training dataset entries from $(x_i, \hat{y}_i, Y_i, s_i)$ to $(x_i, \hat{y}_i, Y_i, s_i, ctx_i)$. Finally, we perform retrieval-augmented fine-tuning (RAFT) (Zhang et al., 2024) of an LLM as the rectifier, using training pairs of the form $(\text{prompt}(x_i, s_i, \bar{y}_i, ctx_i); \hat{y}_i)$, where $\bar{y}_i$ is randomly selected from $K$ responses in $Y_i$.

**Knowledge State Estimator Training.** To improve inference efficiency, we additionally train a knowledge state estimator that directly predicts the LLM's knowledge state $s_i$ for a given question $x_i \in \mathcal{D}$. Since we represent knowledge possession and answer behavior within a knowledge state quadrant, the estimator is formulated as a four-class classification task.

Specifically, given the augmented training dataset $\mathcal{D} = (x_i, \hat{y}_i, Y_i, s_i)_{i=1}^N$ described in § 4.1, we perform supervised fine-tuning of an LLM to serve as the knowledge state estimator, using pairs of the form $(\texttt{prompt}(x_i); s_i)$, where the prompt incorporates the question $x_i$ and the target label is its knowledge state $s_i$. The estimator is parameterized by $\tau$, and its SFT objective is defined as:

$$\mathcal{L}_\tau = -\mathbb{E}_{(x_i,s_i)\sim\mathcal{D}} \left[\log p_\tau(s_i \mid x_i)\right]. \quad (7)$$

This design enables the estimator to obtain a knowledge state in a single forward pass, rather than relying on sampling $K$ responses and computing consistency and semantic entropy. We provide empirical evaluations of the estimator's impact on model performance in § 5.3.

### 4.3 INFERENCE STAGE OF FAITH

We employ the policy model $\pi_\phi$, the estimator model $Est_\tau$, and the RAG model $\pi_{rag}$ to perform factuality-enhanced question answering. Specifically, as shown in Figure 1, Panel (c), given a question $x$, we first predict its knowledge state $s$ in the knowledge state quadrant using the estimator model: $s = \text{Est}_\tau(x)$. We then prompt the policy model $\pi\phi$ with $(x, s)$ to generate the answer $\tilde{y} = \pi_\phi(\text{prompt}(x, s))$. Finally, we apply the RAG model to further rectify the answer produced by the policy model: $\tilde{y}^\star = \pi_{rag}(\text{prompt}(x, s, \tilde{y}, ctx_i))$, obtaining the final answer $\tilde{y}^\star$. We analyze the impact of the RAG model as a rectifier in § 5.3. All prompt templates used during inference are identical to those employed in the training stage.

## 5 EXPERIMENTS

### 5.1 EXPERIMENTAL SETUP

**Datasets.** For training, we adopt the same widely used QA datasets as prior works in FAITH to ensure fair comparison: (1) TriviaQA (Joshi et al., 2017), where questions are from various topics and authored by trivia enthusiasts with evidence documents. (2) SciQ (Welbl et al., 2017b), which focuses on question answering in the scientific domain; and (3) NQ-Open (Kwiatkowski et al., 2019), consisting of Google search queries paired with annotated short-form answers.

For evaluation, we use the test splits of these three datasets as in-domain benchmarks, and employ WebQuestions (Berant et al., 2013a) as an out-of-domain dataset to assess the generalization capability of our approach. Detailed descriptions and statistics of all datasets are provided in Appendix A.4.

---

[1]BGE-base-en-v1.5 : https://huggingface.co/BAAI/bge-base-en-v1.5

Table 2: **Precision and Truthfulness of FAITH (ours)** vs. strong baselines on in-domain (ID) and out-of-domain (OOD) QA datasets. The Average (ID) column denotes the average performance on all three ID datasets. The subscript "sft" denotes ablation results with only supervised fine-tuning (SFT), excluding the PPO and RAG (if applicable) module. Similarly, "sft+ppo" denotes results with SFT and PPO, but excluding the RAG module. All results are reported in percentages.

| Method | TVQA (ID) | | SciQ (ID) | | NQ-Open (ID) | | Average (ID) | | WebQ-QA (OOD) | |
|---|---|---|---|---|---|---|---|---|---|---|
| | Prec. ↑ | Truth. ↑ | Prec. ↑ | Truth. ↑ | Prec. ↑ | Truth. ↑ | Prec. ↑ | Truth. ↑ | Prec. ↑ | Truth. ↑ |
| Llama3-8B | | | | | | | | | | |
| ICL-CoT | 66.68 | 53.37 | 72.34 | 45.90 | 57.34 | 23.60 | 65.45 | 40.95 | 65.97 | 30.85 |
| SFT | 70.80 | 52.57 | 72.18 | 45.40 | 41.41 | 16.57 | 61.46 | 38.18 | 66.46 | 31.18 |
| RL-DPO | 72.08 | 53.96 | 71.23 | 44.20 | 49.65 | 19.18 | 64.32 | 39.11 | 65.99 | 32.41 |
| DTA[2] | 43.99 | 31.73 | – | – | 56.72 | 21.12 | – | – | 61.24 | 30.71 |
| UAlign | 79.14 | 57.04 | 76.44 | 48.00 | 56.60 | 26.09 | 70.72 | 43.71 | 66.88 | 33.01 |
| UAlign$_{sft}$ | 78.76 | 56.68 | 75.87 | 47.65 | 56.02 | 25.49 | 70.22 | 43.27 | 66.12 | 32.58 |
| **FAITH (ours)** | **84.19** | **60.69** | **80.61** | **49.99** | **58.13** | **27.58** | **74.26** | **45.73** | **67.99** | **34.03** |
| **FAITH**$_{sft+ppo}$ | 82.95 | 59.80 | 80.29 | 49.70 | 57.99 | 26.52 | 73.79 | 45.69 | 67.31 | 33.75 |
| **FAITH**$_{sft}$ | 81.24 | 58.77 | 78.85 | 48.65 | 56.72 | 26.21 | 72.27 | 44.54 | 66.94 | 33.10 |
| Mistral-7B-v0.1 | | | | | | | | | | |
| ICL-CoT | 76.73 | 54.78 | 71.87 | 44.20 | 54.47 | 18.22 | 67.69 | 39.06 | 53.43 | 35.76 |
| SFT | 74.57 | 54.77 | 65.85 | 42.50 | 50.82 | 14.42 | 63.74 | 37.08 | 52.24 | 34.33 |
| RL-DPO | 72.20 | 52.98 | 66.44 | 41.80 | 50.95 | 16.42 | 63.19 | 37.06 | 52.01 | 33.87 |
| DTA[2] | 41.33 | 28.78 | – | – | 41.01 | 20.49 | – | – | 56.53 | 23.44 |
| UAlign | 82.10 | 59.05 | 73.21 | 46.70 | 54.17 | 19.64 | 70.82 | 41.79 | 56.47 | 37.02 |
| UAlign$_{sft}$ | 81.07 | 56.47 | 72.45 | 45.87 | 43.32 | 21.55 | 65.61 | 41.30 | 55.34 | 36.87 |
| **FAITH (ours)** | **87.20** | **60.72** | 81.42 | 51.40 | **48.05** | **23.91** | 72.22 | **45.34** | **58.04** | 40.43 |
| **FAITH**$_{sft+ppo}$ | 87.00 | 60.58 | **83.68** | **51.80** | 46.60 | 23.19 | **72.43** | 45.19 | 55.48 | 38.65 |
| **FAITH**$_{sft}$ | 86.51 | 60.24 | 82.88 | 51.30 | 46.16 | 22.96 | 71.85 | 44.83 | 51.99 | **41.77** |

**Evaluation Metrics.**   Consistent with baselines, we employ Precision (*Prec.*) and Truthfulness (*Truth.*) as evaluation metrics. Precision measures the proportion of correctly answered questions among all known questions, reflecting an LLM's ability to accurately articulate its known knowledge. Truthfulness is defined as the proportion of correctly answered known questions plus correctly refused unknown questions over all questions. Further details are provided in Appendix A.5.

**Baselines.**   We evaluate FAITH against five baseline methods that fall into three categories: prompt-based, SFT-based, and RL-based. **(1)** ICL-CoT (Wei et al., 2022): A prompt-based approach that uses few-shot exemplars with reasoning steps to improve answer accuracy. **(2)** Supervised Fine-Tuning (SFT): A standard baseline that fine-tunes LLMs by minimizing the negative log-likelihood of ground-truth answers conditioned on the questions. **(3)** RL-DPO follows Lin et al. (2024) to construct factuality preference dataset to improve the factuality of LLMs by preference optimization. **(4)** Divide-then-Align (DTA): A framework for honest alignment of retrieval-augmented LLMs based on knowledge boundary. It employs multi-objective training that combines DPO loss, SFT loss, and boundary classification loss to align model behavior with knowledge boundary constraints (Sun et al., 2025). **(5)** UAlign leverages uncertainty estimation to elicit LLMs to accurately express factual knowledge that they cannot consistently answer correctly (Xue et al., 2025).

**Training Setup.**   We implement our approach on Llama3-8B (Dubey et al., 2024) and Mistral-7B-v0.1 (Jiang et al., 2023a), applying LoRA for parameter-efficient fine-tuning. Full training details are provided in Appendix A.3.

## 5.2   MAIN RESULTS

We evaluate the effectiveness of our factuality alignment framework FAITH against strong baselines with experimental results presented in Table 2. From the table, we have the following key findings.

**(1) FAITH achieves state-of-the-art performance, outperforming advanced baselines.**   As shown in Table 2, FAITH consistently surpasses five baselines on three in-domain and one out-of-

---

[2]For DTA with Llama3, we directly evaluate the released checkpoint, whereas for DTA with Mistral, we fine-tune Mistral-7B-Instruct on the released training data for Llama3 in a transfer setting. Meanwhile, since DTA requires augmented QA datasets with RAG context and SciQ's augmented version was not released, its results on SciQ are unavailable (denoted as "–" in the table).

domain dataset. For instance, on Llama3-8B model, FAITH achieves an overall precision of 74.26% and truthfulness 45.73% on in-domain datasets, and it attains precision of 67.99% and truthfulness of 34.03% on WebQuestions dataset. We observe similar performance superiority on Mistral-7B model, with the exception of precision on NQ-Open, demonstrating that the effectiveness of FAITH generalizes across models and datasets.

**(2) Natural-language knowledge states are more effective than numerical uncertainty values in guiding knowledge-boundary-aware question answering.** To assess the effectiveness of our knowledge-state-quadrant design, we compare it against numerical uncertainty values. Specifically, we construct a variant of UAlign by eliminating its policy optimization stage and retaining the remaining SFT stage, i.e., we apply SFT with prompts containing numerical uncertainty values. We keep all other settings unchanged. Similarly, we implement FAITH with SFT only, where the model is prompted with natural-language knowledge state drawn from the knowledge-state quadrant. Their performance is reported in Table 2 under **UAlign**$_{sft}$ and **FAITH**$_{sft}$, respectively.

Evaluation shows that replacing numerical uncertainty values with semantically rich knowledge states in natural language yields clear gains in guiding LLMs to understand their knowledge boundary and answer questions accordingly. For instance, on Llama3-8B, FAITH with SFT outperforms UAlign with SFT by 2.05% in precision and 1.27% in truthfulness on average, with even larger improvements observed on Mistral-7B. We attribute these improvements to LLMs' preference for semantically meaningful labels (e.g., "known", "honest") that better convey knowledge boundary. In contrast to fitting abstract numerical values, LLMs more readily interpret and leverage natural language as guidance, enabling knowledge-boundary-aware question answering.

**(3) Reinforcement learning with our proposed reward function improves performance.** We examine the impact of reward function design by comparing the correctness-based binary reward used in UAlign with our fine-grained reward function in Eq. 4. For example, on Llama3-8B, applying PPO with binary reward yields average gains of 0.7% in precision and 0.44% in truthfulness over SFT[3] on three in-domain datasets, whereas FAITH, applying PPO with our reward function, achieves larger improvements of 1.52% in precision and 1.15% in truthfulness, which demonstrates the effectiveness of the fine-grained reward function in incentivizing LLM's generation from both correctness and uncertainty.

**(4) Retrieval-Augmented Fine-Tuning aligns policy model outputs with external knowledge by rectifying potential errors.** As shown in Table 2, comparing the values under **FAITH** with **FAITH**$_{sft+ppo}$, we observe consistent performance improvements across both LLMs, except for SciQ on Mistral-7B. This demonstrates that incorporating external knowledge enhances the truthfulness of LLM's responses. Besides, we manually inspect the corrections made by RAG model to the policy model outputs. Interestingly, some rectifications fail, even altering correct answers into incorrect ones, though such cases are rare. We provide an in-depth analysis of such cases in § 5.3.

## 5.3 ANALYSIS AND DISCUSSION

**Performance of the RAG Model on Post-Hoc Correction to Policy Model Outputs.** For this analysis, we conduct a statistical study on both in-domain (TriviaQA, SciQ, NQ-Open) and out-of-domain (WebQuestions) datasets, with results summarized in Figure 2. Specifically, we examine the responses produced by the policy model that are subsequently modified by the RAG model, and compute the proportion of cases where an incorrect response is corrected into a correct one versus the reverse. We find that the proportion of correct rectifications consistently exceeds that of erroneous

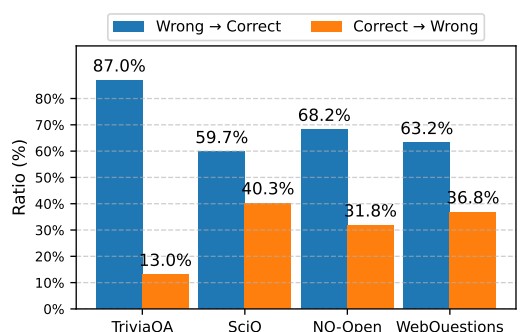

Figure 2: Ratios of on four datasets.

rectifications across all datasets. Notably, on TriviaQA, 87% of the policy model outputs modified by the RAG model are corrected successfully, demonstrating that incorporating external knowledge effectively compensates for the insufficiency of relying solely on internal knowledge.

---

[3]Calculated as the difference between the metric values reported under **UAlign** and **UAlign**$_{sft}$ in Table 2

**Calculate Knowledge State: Estimator vs. Sample $K$ Responses.** We further investigate the impact of different knowledge state estimation strategies on model performance during inference, comparing model-based estimation and sampling-based estimation. The model-based approach corresponds to our trained estimator model, whereas the sampling-based approach follows a two-stage pipeline: first sampling $K$ responses using prompts with different one-shot examples, and then computing the knowledge state with Eq. 3. For this analysis, we set $K = 6$, consistent with the training stage. We present the evaluation results for precision and truthfulness in Table 3, and we observe that sampling-based estimation yields slightly higher precision and truthfulness in most cases.

These findings indicate the distribution of the knowledge states can be captured by a trained LLM, while also highlighting a trade-off between efficiency and performance: sampling provides better performance with interpretable uncertainty measures, whereas model-based estimation avoids $K$ rounds of inference with only minimal performance degradation.

Table 3: Performance comparison between model-based and sampling-based knowledge state estimation ($K = 6$). Results are reported on precision and truthfulness across all datasets and models.

| Method | TVQA (ID) | | SciQ (ID) | | NQ-Open (ID) | | Average (ID) | | WebQ-QA (OOD) | |
|---|---|---|---|---|---|---|---|---|---|---|
| | *Prec.* ↑ | *Truth.* ↑ | *Prec.* ↑ | *Truth.* ↑ | *Prec.* ↑ | *Truth.* ↑ | *Prec.* ↑ | *Truth.* ↑ | *Prec.* ↑ | *Truth.* ↑ |
| | | | | | Llama3-8B | | | | | |
| **Estimator** | 82.95 | 59.80 | 80.29 | 49.70 | 57.99 | 26.52 | 73.79 | 45.69 | 67.31 | 33.75 |
| **Sample-based** | **83.99** | **59.86** | **83.20** | **51.50** | **58.23** | **26.93** | **75.14** | **46.10** | **67.85** | **34.07** |
| | | | | | Mistral-7B-v0.1 | | | | | |
| **Estimator** | 87.00 | 60.58 | 83.68 | 51.80 | **46.60** | **23.19** | **72.43** | 45.19 | 55.48 | 38.65 |
| **Sample-based** | **87.43** | **60.88** | **84.01** | **52.00** | 45.66 | 22.71 | 72.37 | **45.20** | **55.70** | **38.80** |

**Training-time Scaling: the Impact of Number of Sampled Responses.** We study the training-time scaling behavior, i.e., how the number of sampled responses $K$ used during data augmentation influences the training performance of the policy model. The default $K$ we use is 6 in the main framework for efficiency. Here, specifically, we increase $K$ from 6 to 8, 10, and 12

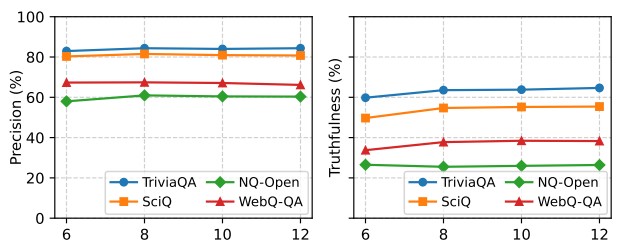

Figure 3: Training-time scaling with different numbers of sampled responses ($K$).

during data augmentation, resulting in augmented datasets that differ only in the values of $K$. These datasets are then used to train both estimator and policy model (including reference model and reward model). Finally, we evaluate the trained models and compare the performance. As shown in Figure 3, increasing $K$ beyond 6 does not yield noticeable improvements in either precision or truthfulness. This suggests that sampling $K = 6$ responses during data augmentation is already sufficient and effective to capture the distribution of the model's knowledge state, while also keeping the efficiency. In other words, while larger $K$ values slightly expand the coverage of sampled responses, they do not translate into significant gains in downstream performance, indicating minor effects. The detailed numerical results are provided in Table 4 in the Appendix A.6.

Finally, we present case studies and discuss the limitations in Appendix A.7 and A.8, respectively. For reproducibility, the code is available in an anonymous repository at `https://anonymous.4open.science/r/FAITH-33A3`

## 6 CONCLUSION

We present a post-training framework, called FAITH, for factuality alignment in LLMs. Our approach estimates uncertainty and translates the numerical values into natural-language knowledge states that measure the knowledge possession and answering behavior of LLMs. Meanwhile, we design a fine-grained reward function to incentivize both correctness and uncertainty of LLM's response. Finally, we introduce a trained RAG model to rectify potentially incorrect responses generated by policy model. Experiments show that FAITH substantially outperforms recent baselines in truthfulness and precision. We hope this work contributes to building more faithful and factual LLMs as part of the broader community effort.

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

# A APPENDIX

## A.1 DERIVATION OF RULES FOR MAPPING UNCERTAINTY VALUES TO KNOWLEDGE STATES

The proposed rule-based mapping from uncertainty values ($Consistency$ and $SE$) to natural-language knowledge states is as follow:

Let $Y_i = \{y_i^k\}_{k=1}^K$ be K sampled responses from model and $\hat{y}$ be the reference answer. *Knowledge possession* is captured by $Consistency$, where:

- Consistency $> 0$ indicates there exists at least one response $y_i^k$ matches the ground-truth answer $\hat{y}$, meaning the model possesses the required knowledge with a very high probability.

- Consistency $= 0$ represents that the model fails to answer the question correctly with $K$ times, indicating the model does not possess relevant knowledge to the question. For a given question, we assume that if the LLM possesses the relevant knowledge, the probability of answering it correctly is $p_\theta = 0.5$. We set the confidence level to $\alpha = 0.05$. If none of the $K$ sampled responses are correct, then with confidence $1 - \alpha$ we can reject the hypothesis that the model's probability of producing a correct answer satisfies $p_\theta \geq 0.5$. Under the assumption $p_\theta = 0.5$, sampling $K = 6$ responses is sufficient to show that if none of them are correct, the model is unlikely to possess the relevant knowledge.

*Answer Behavior* is measured by *semantic entropy*:

- The magnitude of semantic entropy reflects the model's uncertainty at the semantic level: a higher value indicates diverse or conflicting semantic outputs (greater ambiguity), while a lower value suggests more consistent and deterministic semantic interpretations.

- Semantic entropy equals zero when all generated outputs are semantically equivalent, i.e., they fall into the same semantic cluster with no competing interpretations. From the semantic perspective, the model is completely certain, exhibiting neither ambiguity nor polysemy.

Given these interpretations, the mapping rules follow a logically consistent decision path:

1. If Consistency $> 0$ and SE $= 0$, the model is judged to possess the relevant knowledge of a question and honestly provides consistent correct responses, corresponding to the knowledge state **KH**.

2. If Consistency $> 0$ and SE $\neq 0$, the model produces a mix of correct and incorrect answers, indicating insufficient mastery of the knowledge to express it accurately. The reason for this gap could be decoding strategy, hallucination snowballing, misalignment issues (Liang et al., 2024). This corresponds to the knowledge state **K¬H**.

3. If Consistency $= 0$ and SE $= 0$, the model lacks correct knowledge but converges on a single interpretation, corresponding to the knowledge state **¬KH**.

4. In all other cases, the knowledge state is classified as **¬K¬H**.

Overall, the mapping is determined by two factors:

$$\underbrace{\text{Knowledge possession (Consistency)}}_{\text{know}} \quad \text{and} \quad \underbrace{\text{Answer honesty (Semantic Entropy)}}_{\text{tell}},$$

which together define a quadrant of four cognitive states, ensuring both interpretability and completeness.

## A.2 PROMPT TEMPLATES

We illustrate the prompt templates used in this work in Figure 4, detailing the input structure, incorporated knowledge states, and output format. The templates explicitly define how a question is combined with its corresponding knowledge state, optionally with retrieved external context, and then formatted to elicit model responses. By making this structure explicit, the figure clarifies how prompts guide the model during both training and inference, ensuring consistency across stages.

Moreover, the design rationale highlights how natural-language descriptions of knowledge states are integrated into the prompt, which is essential for conveying uncertainty information in a semantically interpretable way.

---

**Prompt template for sampling responses**

You are an excellent Question-Answering assistant. Please answer the following question based on your knowledge.

### Question ###: {demo_question_1}
### Answer ###: {demo_answer_1}

### Question ###: {input_question}
### Answer ###:

---

**Prompt template for supervised fine-tuning of the reference model**

You are an excellent Question-Answering assistant. Please answer the following question based on your knowledge.

### Question ###: {THE QUESTION FROM DATASET}

### Self-Eval ###: {THE KNOWLEDGE STATE FROM DATASET}

### Output ###: {GOLDEN ANSWER}

---

**Prompt template for policy model optimization**

You are an excellent Question-Answering assistant. Please answer the following question based on your knowledge.

### Question ###: {THE QUESTION FROM DATASET}

### Self-Eval ###: {THE KNOWLEDGE STATE FROM DATASET}

### Answer ###: {GOLDEN ANSWER}

---

**Prompt template for RAFT a RAG model**

You are an excellent Question-Answering assistant. Please answer the following question based on your knowledge.

### Question ###: {THE QUESTION FROM DATASET}

### Self-Eval ###: {THE KNOWLEDGE STATE FROM DATASET}

### Prior Judgment ###: {RANDOMLY SELECTED RESPONSE FROM $Y_i$}

### Retrieve Documents ###: related passages: ###passage 1###;###passage 2###;###passage 3###

### Posterior Answer ###: {GOLDEN ANSWER}

---

**Prompt template for RAG model in inference**

You are an excellent Question-Answering assistant. Please answer the following question based on your knowledge.

### Question ###: {THE QUESTION FROM DATASET}

### Self-Eval ###: {THE KNOWLEDGE STATE FROM DATASET}

### Prior Judgment ###: {POLICY MODEL'S OUTPUT}

### Retrieve Documents ###: related passages: ###passage 1###;###passage 2###;###passage 3###

### Posterior Answer ###: {GOLDEN ANSWER}

---

**Prompt template for supervised fine-tuning of the estimator model.**

You are an excellent Question-Answering assistant. Please answer the following question based on your knowledge.

### Question ###: {THE QUESTION FROM DATASET}

### Self-Eval ###: {THE KNOWLEDGE STATE FROM DATASET}

---

Figure 4: All the prompt templates employed in FAITH (our framework).

### A.3  DETAILS OF TRAINING

All experiments are conducted on a cluster equipped with 4 × NVIDIA A40 and/or 4 × NVIDIA 4090D GPUs.

For supervised fine-tuning (SFT) of both the reference model and estimator in FAITH, we train for 3 epochs. We adopt the Adam optimizer with an initial learning rate of 2e-4. We apply LoRA with a rank of 32, alpha of 16, and a dropout rate of 0.05, targeting all layers. The batch size per device is set to 8, with gradient accumulation steps of 8, leading to a total batch size of 256. The learning rate scheduler follows a cosine decay with a warmup ratio of 0.0.

For policy optimization, both the reward model (RM) and the PPO stages are trained for 2 epochs. We adopt the Adam optimizer with an initial learning rate of 1e-5. LoRA is applied with a rank of 8, alpha of 16, and a dropout rate of 0.05, again targeting all layers. The per-device batch size is set to 4, with gradient accumulation steps of 8, leading to a total batch size of 128. The learning rate scheduler is cosine decay, and the warmup ratio is 0.0, consistent with the SFT stage.

### A.4  DETAILS OF DATASET

**SciQ:**  The SciQ dataset (Welbl et al., 2017a) contains 13,679 crowdsourced science examination questions covering subjects such as physics, chemistry, and biology. Although originally released in multiple-choice format, in our setting all answer options are removed, and each question is reformulated as an open-ended query requiring a direct answer. For most questions, an accompanying paragraph with supporting evidence is provided, offering factual context that can be utilized to guide answer generation and factual alignment. In our experiments, 11,679 samples are used for training and 1,000 samples are reserved for validation, with the remaining questions serving as an in-domain test set.

**TriviaQA:**  TriviaQA (Joshi et al., 2017) is a large-scale reading comprehension dataset containing over 650K question-answer-evidence triples, with questions authored by trivia enthusiasts and evidence documents collected from Wikipedia and the web. In our work, for constructing the augmented dataset, we pre-process and sample half of the original training set.

**NQ-Open:**  NQ-Open (Kwiatkowski et al., 2019) is an open-domain QA benchmark derived from the Natural Questions dataset, where real user queries are paired with English Wikipedia passages as the knowledge source. In our work, we employ NQ-Open for augmented dataset construction. Similarly, to ensure fair comparison and reduce computational cost, we sample half of the original training data.

**Web-Questions:**  The WebQuestions dataset (Berant et al., 2013b) comprises 6,642 question–answer pairs, where each question can be answered using Freebase, a large-scale knowledge graph. The majority of questions are centered around a single named entity and reflect typical queries collected from the web around 2013. In our experiments, we only employ its test set (1348 item) for the evaluation under the out-of-domain evaluation setting.

### A.5  DETAILS OF EVALUATION METRICS

Truthfulness quantifies the proportion of correct responses among all provided answers, reflecting the LLM's overall reliability in expressing knowledge. The formula for Truthfulness is given as follows:

$$\text{Truthfulness} = \frac{\text{UR} + \text{KC}}{\text{KC} + \text{KI} + \text{KR} + \text{UC} + \text{UI} + \text{UR}} \tag{8}$$

Precision measures the proportion of correctly answered questions among those for which the model possesses the relevant knowledge, reflecting the LLM's ability to accurately convey known facts. The formula for Precision is given as follows:

$$\text{Precision} = \frac{\text{KC}}{\text{KI} + \text{KC} + \text{KR}} \tag{9}$$

## A.6 NUMERICAL RESULTS OF TRAINING-TIME SCALING

Table 4 reports the detailed numerical results corresponding to the training-time scaling analysis. The table compares precision and truthfulness across different values of $K$ (6, 8, 10, 12). Consistent with Figure 3, the results show that increasing $K$ beyond 6 does not yield noticeable gains, confirming that $K = 6$ is sufficient to capture the model's knowledge state distribution during data augmentation.

Table 4: Training-time scaling with different numbers of sampled responses ($K$) on Llama-3-8B.

| # Responses | TVQA (ID) | | SciQ (ID) | | NQ-Open (ID) | | Average (ID) | | WebQ-QA (OOD) | |
|---|---|---|---|---|---|---|---|---|---|---|
| | Prec. ↑ | Truth. ↑ | Prec. ↑ | Truth. ↑ | Prec. ↑ | Truth. ↑ | Prec. ↑ | Truth. ↑ | Prec. ↑ | Truth. ↑ |
| Llama-3-8B | | | | | | | | | | |
| K=6 | 82.95 | 59.80 | 80.29 | 49.70 | 57.99 | 26.52 | 73.79 | 45.69 | 67.31 | 33.75 |
| K=8 | 84.36 | 63.55 | 81.52 | 54.70 | 60.95 | 25.51 | 75.61 | 47.92 | 67.42 | 37.76 |
| K=10 | 84.01 | 63.79 | 80.94 | 55.20 | 60.44 | 25.98 | 75.13 | 48.32 | 67.10 | 38.43 |
| K=12 | 84.36 | 64.65 | 80.76 | 55.40 | 60.35 | 26.40 | 75.16 | 48.82 | 66.15 | 38.28 |

## A.7 CASE STUDY

In our method FAITH, one focus is to train RAG model to align the policy model's output with external knowledge. The RAG model is provided with retrieved passages as context, allowing it to rectify or retain the policy model's responses. In this section, we analyze three types of corrections, with representative cases shown in Table 5, 6, and 7, as case studies. Specifically, the three correction types are summarized as follows:

1. **Implicitly Supported Correction**: The initial answer from the policy model was incorrect, but after applying our trained RAG model, the final answer was corrected. Notably, the retrieved passages did not verbatim reproduce the correct answer, but contained key information or semantic cues related to the correct answer. Details can be found in Table 5.

2. **Explicitly Supported Correction**: The policy model initially produced an incorrect output, but after applying our trained RAG model, the final output was corrected. In this process, the retrieved content from RAG not only directly reproduced the correct answer but also provided additional information related to it, thereby supporting the model's correction. Details can be found in Table 6.

3. **Misleading Override**: The policy model initially produced the correct answer. However, after applying our trained RAG model, the output was incorrectly altered. This occurred because the retrieved content contained misleading information that contradicted the correct answer, ultimately leading to an erroneous output. Details can be found in Table 7.

Table 5: Case Study Analysis 1 of Model Responses. Specifically, **A** denotes Ground Truth, **A1** denotes the policy model's output, and **A2** denotes the final output.

| Type | Question & Answer | Retrieved Passages | Our Analysis |
|------|-------------------|--------------------|--------------|
| **Implicitly Supported Correction** | **Q:** Protists play critically important ecological roles as producers and, on the other end of food webs, as what? 
 **A:** decomposers 
 **A1:** consumers 
 **A2:** decomposers | **I.** In real world ecosystems, there is more than one food chain for most organisms, since most organisms eat more than one kind of food or are eaten by more than one type of predator. A diagram that sets out the intricate network of intersecting and overlapping food chains for an ecosystem is called its food web. Decomposers are often left off food webs, but if included, they mark the end of a food chain. **Thus, food chains start with primary producers and end with decay and decomposers.** 
 **II.** Food webs have trophic levels and positions. Basal species, such as plants, form the first level and are the resource-limited species that feed on no other living creature in the web. Basal species can be autotrophs or detritivores, including decomposing organic material and its associated microorganisms, which we defined as detritus, micro-inorganic material, and associated microorganisms (MIP), and vascular plant material. 
 **III.** The microbial food web refers to the combined trophic interactions among microbes in aquatic environments. These microbes include viruses, bacteria, algae, and heterotrophic protists (such as ciliates and flagellates). In aquatic environments, microbes constitute the base of the food web. Single-celled photosynthetic organisms such as diatoms and cyanobacteria are generally the most important primary producers in the open ocean. Many of these cells, especially cyanobacteria, are too small to be captured and consumed by small crustaceans and planktonic larvae. Instead, these cells are consumed by phagotrophic protists, which are readily consumed by larger organisms. | Some protists do function as A1 (consumers), but the model fails to accurately address the specific context of "the other end of food webs" posed in the question. This suggests the model's insufficiency in effectively utilizing its internal knowledge to answer the question, or a failure to correctly understand the question, particularly its key constraints. After introducing the retrieved information to our trained RAG model, it successfully corrected the answer to A2 (decomposers). The retrieved information, in paragraph I, provides the crucial background knowledge: "Thus food chains start with primary producers and end with decay and decomposers." This information does not explicitly state that "protists are decomposers." Instead, it requires the model to synthesize this information with "protists" and "the other end of food webs" to deduce the correct answer. We denote this process as **Implicitly Supported Correction**. |

Table 6: Case Study Analysis 2 of Model Responses. Specifically, **A** denotes Ground Truth, **A1** denotes the policy model's output, and **A2** denotes the final output.

| Type | Question & Answer | Retrieved Passages | Our Analysis |
|---|---|---|---|
| **Explicitly Supported Correction** | **Q:** Rita Coolidge sang the title song for which Bond film? **A:** Octopussy **A1:** North by Northwest **A2:** Octopussy | **I.** Octopussy is the soundtrack for the eponymous thirteenth James Bond film. The score was composed by John Barry, the lyrics by Tim Rice. **The opening theme, All Time High is sung by Rita Coolidge and is one of six Bond film title songs or songs that are not named after the film's title.** The original compact disc released in 1985 by A&M Records, was recalled because of a printing error and became a rarity. **II.** Another Way to Die is a song by American musicians Jack White and Alicia Keys. Written and produced by White as the theme song to the 2008 James Bond film Quantum of Solace, it was released as a single in the United States on September 30, 2008, and in Europe on October 20, 2008. **III.** Tomorrow Never Dies is the song, performed by Sheryl Crow, which served as the theme song to the James Bond film of the same name. The song was co-written by Crow and the song's producer Mitchell Froom, and became her fifth UK Top 20 hit, peaking at No. 12 in 1997. Another song, Tomorrow Never Dies, written by the movie's composer David Arnold and performed by k.d. lang, was originally produced as the official theme tune. When Crow's song became the official theme, the k.d. lang song was relegated to the end credits, and renamed Surrender. | The model's initial response, A1 (North by Northwest), is a significant factual error, as this film is not even part of the James Bond series. This indicates a substantial knowledge gap or a "hallucination" in the model's internal knowledge base. After the RAG intervention, the model successfully corrected the answer to A2 (Octopussy). The retrieved information in paragraph I contains all the key details required to rectify the error. The passage explicitly states, "The opening theme, All Time High is sung by Rita Coolidge" and that the soundtrack was for the film Octopussy. The model simply needed to match the key entity from the question, "Rita Coolidge", with the retrieved text to directly find the name of the film for which she sang the theme. The entire process involves direct information extraction and localization, requiring almost no complex reasoning. We denote this process as **Explicitly Supported Correction**. |

Table 7: Case Study Analysis 3 of Model Responses. Specifically, **A** denotes Ground Truth, **A1** denotes the policy model's output, and **A2** denotes the final output.

| Type | Question & Answer | Retrieved Passages | Our Analysis |
|---|---|---|---|
| **Misleading Override** | **Q:** Which grand slam did Pete Sampras not win in the 20th century? 
 **A:** French 
 **A1:** French Open 
 **A2:** Wimbledon | **I.** As the Swiss national anthem played Federer was overcome with emotion after finally capturing the elusive title at Roland Garros. This match was momentous in the history of tennis. After missing the chance to equal Pete Sampras' then-record of fourteen Grand Slam championships of all time when he lost to Rafael Nadal in the final of the Australian Open earlier in the year, Federer finally did so by winning the French Open for the first time. Sampras himself commented on Federer following the victory saying, Regardless he [Federer] goes down as the greatest ever. 
 **II.** In the eight Wimbledons inclusive between 1993 and 2000, 1996 was the only year that Sampras would fail to win the championship at Wimbledon. Sampras lost in the quarterfinals of Wimbledon to the eventual winner, Richard Krajicek, the tournament's 17th-seed. The match lasted three long sets, with Krajicek winning 7-5, 7-6, 6-4. In the quarterfinals of the US Open, Sampras vomited on the court at 1–1 in the final set tiebreak (due to dehydration) while facing Àlex Corretja; nonetheless, Sampras would win that match. 
 **III.** He beat former champion Michael Stich in the fourth round and met Sampras in the quarterfinals. By that time, he had managed to turn his notably weak slice backhand into an aggressive topspin shot. Krajicek shocked the tennis world by defeating Sampras in straight sets, becoming the only player to beat Sampras in a Wimbledon singles match in the eight-year period from 1993 until Sampras' fourth-round loss to Roger Federer in the 2001 tournament. | This is a failure case of a 'correct-to-incorrect' reversal caused by the RAG. The model's initial judgment, A1 (French Open), was correct, indicating that its internal knowledge base already contained the key fact about Sampras's career. However, the intervention of RAG instead led to a degradation in performance. The core of the failure lies in the Retrieval stage. The retrieved information, though related to the key entities "Pete Sampras" and "Grand Slam", did not align with the question's specific requirement ("did not win" in the 20th century). The retrieved content, particularly in paragraphs II and III, repeatedly and in detail described a specific loss Sampras had at Wimbledon (in 1996 to Krajicek). Phrases like "fail to win the championship at Wimbledon" became a strong and irrelevant distracting signal. When generating the final answer, the model over-relied on this incorrectly retrieved and distracting content, thereby ignoring its own correct prior knowledge. It was misled into outputting the incorrect answer A2 (Wimbledon). We denite this process as **Misleading Override**. |

### A.8 LIMITATIONS

**Reward Function Design.** Our reward function is derived from heuristic rules that are straightforward to formulate and intuitively easy to interpret. In practice, we observe that this design works well empirically and provides meaningful guidance for aligning model behavior. However, the current formulation lacks rigorous theoretical guarantees, leaving room for future work to establish a stronger theoretical foundation for its effectiveness.

**Computational Overhead.** During dataset construction, we sample $K$ responses and build a vector database. At inference time, our pipeline first uses $Est_\tau$ to estimate the knowledge state $s$, then applies the policy model $\pi_\phi$ to generate an answer, and finally employs $\pi_{rag}$ for rectification. Even without rectification, two model inferences are required, rather than a single end-to-end pass. For future work, we plan to explore more efficient approaches for cognitive-state estimation, such as lightweight estimators derived from LLM internal representations (Zhu et al., 2025)[4].

**Unexplored Aspects of RAG Effectiveness.** Our current study does not investigate how the quality of the data used to build the vector database affects FAITH's performance. For example, on the SciQ dataset with Mistral-7B, incorporating external knowledge does not improve correction effectiveness, which may be related to the quality of the retrieved context. In addition, we have not explored more effective ways of leveraging external knowledge for rectification, such as integrating RAFT directly the SFT stage and accordingly applying PPO training on top of the RAFT-enhanced model, rather than additionally train a RAG model.

### A.9 REPRODUCIBILITY STATEMENT

We have made several efforts to ensure the reproducibility of our work. We provided detailed descriptions of the datasets used in our experiments, all of which are publicly available. Our method is thoroughly explained in dedicated sections § 4, and we also provide detailed training parameters § 5.1. Finally, we have submitted the code via this anonymous repository at `https://anonymous.4open.science/r/FAITH-33A3`

We hope that these measures will facilitate the replication of our work by other researchers and further advance the field.

### A.10 THE USE OF LARGE LANGUAGE MODEL

The authors acknowledge the use of OpenAI ChatGPT solely for enhancing the coherence of the final manuscript, and providing assistance with coding for data processing.

---

[4]This related work has been publicly available on arXiv since September 16, 2025, a week before the ICLR paper submission deadline.

