# OpenReview forum: "FAITH: Factuality Alignment through Integrating Trustworthiness and Honestness"
_ICLR.cc/2026/Conference — ICLR 2026 Conference Withdrawn Submission_

### Official Review · Reviewer_WYeg · 2025-10-30

**Soundness:** 2
**Presentation:** 3
**Contribution:** 2
**Rating:** 4
**Confidence:** 3

**Summary:**

The paper proposes FAITH, a post-training framework to enhance the factual accuracy and truthfulness of Large Language Models (LLMs) by integrating natural-language uncertainty signals (knowledge states) with external knowledge (RAG). While the core idea of mapping numerical uncertainty into semantically rich knowledge states for alignment is novel and shows promising empirical results, the paper suffers from several significant methodological, theoretical, and engineering shortcomings that undermine its contribution and rigor.

**Strengths:**

1. The study introduces the novelty of mapping traditional numerical uncertainty metrics, like Consistency scores and Semantic Entropy, into a four-quadrant natural-language knowledge state.

2. FAITH designs a fine-grained reward function ($R_{FAITH}$) that explicitly considers both the correctness of the response and the uncertainty (expressed via the knowledge state). FAITH could train RAG models to leverage external knowledge (retrieved from a Wikipedia vector database) to rectify potentially incorrect answers generated by the policy model

3. FAITH consistently surpasses five strong and recent baselines across four knowledge-intensive benchmarks (three in-domain and one out-of-domain), showing its effectiveness.

**Weaknesses:**

1. The proposed fine-grained reward function, $R_{FAITH}$, is based on heuristic rules (e.g., $+2$ for Known and Honest, $-2$ for Not Known and Not Honest). But why UnKnown and Honest -1 in the Figure 1.

2. Defining "honestness" simply as semantic entropy ($\text{SE}$) equal to zero is conceptually flawed. Semantic entropy of zero means all sampled responses are semantically equivalent. When a model does not possess the relevant knowledge ($\text{Consistency}=0$) but still outputs a semantically consistent, incorrect answer ($\text{SE}=0$), this state ($\neg\text{KH}$) represents confident hallucination. Labeling this most dangerous type of factual error as "Honest" contradicts the goal of factuality alignment and human intuition about honesty.

3. The criterion for "Have knowledge" is based purely on $\text{Consistency} > 0$, meaning merely "at least one response... matches the ground-truth answer" out of $K$ samples. What is the difference between it and the correctness of y? It seems not to encourage the model to output a consistent answer.

4. More settings of the number of sampled responses should be given, especially the numbers below 6. If the performance is not strongly affected by the samples, why not use less, how about using only 1 response?

**Questions:**

See weakness.

---

### Official Review · Reviewer_Wp3H · 2025-10-31

**Soundness:** 2
**Presentation:** 3
**Contribution:** 3
**Rating:** 4
**Confidence:** 3

**Summary:**

FAITH is a post-training method that makes LLMs answer when they know and refuse when they don’t. It derives textual knowledge-states from closed-book consistency/semantic uncertainty, trains a PPO policy with a multi-level reward to favor correct answers or refusals accordingly, and uses a lightweight estimator to predict the state at test time. A RAFT-style RAG rectifier optionally edits outputs with top-k Wikipedia passages. Across QA benchmarks, FAITH boosts precision and truthfulness (correct answers + correct refusals), with RAG helping partially.

**Strengths:**

1. The paper tackles an important problem: enabling LLMs to be trustworthy by answering when they know and refusing when they don’t.
2. The integration of truthfulness and honesty as interpretable knowledge states to guide RL training is novel and well-motivated.
3. The writing is clear and well-structured, and figures are easy to follow. The proposed improvements over baselines are non-trivial and well-demonstrated.

**Weaknesses:**

1. **Baselines and fairness:**
- Most baselines do not have access to external retrieval, while FAITH uses RAG. This raises fairness concerns in direct comparison.
- It would strengthen the work to rag relevant baselines liek RAG-SFT, or at least clarify whether RAG alone offers similar improvements.

2. **Limited analysis on knowledge states:**
- The knowledge state definition is central to the paper, but there is little analysis of its behavior. How do the distributions of knowledge states change before and after training?
- Since the knowledge estimator is trained off-policy, does the policy’s post-training behavior drift away from the estimator’s assumptions, potentially reducing its accuracy?
- It would be useful to visualize or quantify how the RL process reinforces or shifts certain states (e.g., more “known but refused” vs. “unknown but answered”).

3. **RAG effectiveness and error analysis:**
- The FAITH with RAG system shows only marginal improvements over without RAG. Is this due to retrieval noise or the rectifier model being too conservative?
- The authors should report what fraction of queries are actually modified by the rectifier. While the paper notes a high correction accuracy (e.g., 87% of changed answers are correct on triviaqa), the overall gain seems small—perhaps because only a small subset is affected.

**Questions:**

1. The reward definition in Line 254 does not fully align with Figure 1; please clarify or fix the inconsistency.
2. An ablation comparing rule-based vs. learned reward would clarify whether training a reward model instead of directly apply the designed rule truly adds value beyond smoothing or efficiency.

---

### Official Review · Reviewer_yKKW · 2025-10-31

**Soundness:** 3
**Presentation:** 2
**Contribution:** 2
**Rating:** 2
**Confidence:** 5

**Summary:**

This paper proposes FAITH, a post-training framework to improve the factuality of Large Language Models (LLMs) and address the "know-tell gap." The core idea is to characterize an LLM's internal state using two metrics calculated from $K$ sampled responses: Consistency (accuracy against a golden answer) and Semantic Entropy (semantic dispersion of answers). These two numerical values are then mapped to a four-part "knowledge state quadrant" (e.g., "Known and Honest," "Known and not Honest") expressed in natural language.

The framework involves several training stages:

1. Data Augmentation: Creating the knowledge state labels for the training data.
2. SFT: Fine-tuning a reference model to answer questions given the {question, knowledge_state} prompt.
3. Reward Modeling: Training a reward model (RM) based on a fine-grained reward that combines correctness and the knowledge state.
4. Policy Optimization: Fine-tuning a policy model using PPO with the trained RM.
5. RAG Model: Separately training a RAG model to "rectify" the policy model's output.
6. Estimator Model: Training a classifier to predict the knowledge state at inference time to avoid the cost of $K$ samples.

Experiments on Llama3-8B and Mistral-7B-v0.1 across four QA benchmarks (three in-domain, one out-of-domain) show that the full FAITH framework outperforms several baselines, including SFT, DPO, and UAlign.

**Strengths:**

The paper tackles the critical and well-recognized problem of LLM factuality and the "know-tell gap". The central proposal to move beyond simple numerical uncertainty scores (like in UAlign) to a richer, semantically meaningful "knowledge state" expressed in natural language is intuitive. LLMs are well-suited to understanding and conditioning on such natural language prompts. The results are overall positive and easy to follow.

**Weaknesses:**

- My main criticism of this work is that it presents a complex, multi-stage framework by gluing together several established components (SFT, PPO, RAG), but the empirical evidence suggests the primary contribution and the vast majority of the performance gain come from the SFT stage alone. The subsequent "core" components add significant complexity for what appears to be only marginal benefit. The ablation study in Table 2 is very telling. The $FAITH_{sft}$ model (which only uses the novel data augmentation and SFT) appears to do all the heavy lifting. This undermines the paper's narrative that the PPO and RAG components are critical contributions. The synergy between these components is not demonstrated; they just look like complex, low-impact additions.
- The entire "knowledge state quadrant" mapping hinges on the definitions of Consistency and Semantic Entropy. The definition for "knowledge possession" (Consistency > 0) is based on a statistical argument that itself rests on a completely arbitrary and unsubstantiated assumption: that the probability of a correct answer $p_\theta$ for a "known" fact is 0.5. This 50% threshold is not justified and appears to be a value of convenience. Though Figure 3 shows no improvement with larger K, it's an indirect measurement. The scaling curve for K from 1-6 is also missing.
- The framework estimates the knowledge state $s_i$ once from the base reference model before policy optimization. The PPO model $\pi_\phi$ is then trained to align with this static state. However, the entire point of PPO is to change the model's policy and (by extension) its expression of knowledge. This creates a fundamental mismatch: the policy model is being optimized against a "knowledge state" that may become stale or inaccurate as the model itself improves. The paper does not address this potential misalignment.

**Questions:**

- The authors use the non-standard term "Reference Model Training" to describe what is universally known in RLHF literature as Supervised Fine-Tuning (SFT). This is confusing, as the "reference model" is the role this frozen SFT model plays during the PPO stage, not the name of the training process itself. I strongly recommend replacing this term with "Supervised Fine-Tuning (SFT)" throughout the paper for clarity.
- The introduction (lines 47) and the section on semantic entropy (line 339) vaguely critique "prior approaches" without sufficient specificity. Which specific prior works "lack semantic richness" or are "affected by response length"? The related work critique should be more targeted.
- The proposed reward $R_{FAITH}$ is a sum of $R_{\text{correctness}}$ and a deterministic lookup $R_{\text{uncertainty}}$ based on the knowledge state $s_i$. Why train the reward model $RM_\theta$ to predict this combined value? A simpler approach would be to train $RM_\theta$ to predict only $R_{\text{correctness}}$ and then add the deterministic $R_{\text{uncertainty}}$ value post-hoc during PPO. Was this simpler, more direct approach evaluated?
- What is the 4-class classification accuracy of the knowledge state estimator against the labels generated during data augmentation?
- What is the accuracy of the Reward Model ($RM_\theta$) in predicting the fine-grained reward scores?
- Detailed training/test set size of each dataset, how's the checkpoint selected, etc.
- The reference section of ICLR papers generally does not explicitly include editors' names for conference papers.

---

### Official Review · Reviewer_JrZG · 2025-11-02

**Soundness:** 2
**Presentation:** 3
**Contribution:** 3
**Rating:** 2
**Confidence:** 4

**Summary:**

This paper introduces post-training techniques for improving LLMs on question answering tasks. The basic idea to incorporate signals about model uncertainty and confidence into the input prompts (determined by sampling responses), and training the model to encourage responses consistent with their internal "knowledge state". Results suggest that this improves accuracy over a number of benchmarks.

**Strengths:**

- The high level idea of making the model more aware of its knowledge state and training it to be consistent with that makes sense.
- The results are somewhat strong, showing improvements over a number of baselines on a number of benchmarks (though some caveats noted below).

**Weaknesses:**

- While the high-level idea is clear, the details of why the proposed methods help with factuality are not clear. E.g., the reward in Eq (4) will have the same value of R_uncertainty, irrespective of the sampled answer. So why does adding a constant to the reward help here?
- RAG is incorporated to refine the answers predicted by the model, but this seems orthogonal to the basic motivation of improving closed-book factuality. Given RAG is used, the paper should be comparing to baselines which also use RAG, as this can significantly enhance the accuracy of the base model.
- A knowledge state estimator is used for predicting how confident the model would be on test-time questions. While this seems to be effective on the datasets studied, its not clear to me how this can work more generally -- given a new fact at test time its impossible to determine a-priori whether this would be known to the target model or not.
- The overall idea is very similar to the UAlign paper [1], which is used as a baseline. Beyond using natural language expressions for confidence and uncertainty, I would like to how else the paper differs from the UAlign approach. This is not clearly described in the paper.

[1] https://arxiv.org/abs/2412.11803

**Questions:**

- In Figure 3, how do the trends look like for K < 6?

---

### Note · Authors · 2025-12-23

I have read and agree with the venue's withdrawal policy on behalf of myself and my co-authors.